# The Effects of Sous Vide, Microwave Cooking, and Stewing on Some Quality Criteria of Goose Meat

**DOI:** 10.3390/foods12010129

**Published:** 2022-12-27

**Authors:** Monika Wereńska, Gabriela Haraf, Andrzej Okruszek, Weronika Marcinkowska, Janina Wołoszyn

**Affiliations:** Department of Food Technology and Nutrition, Wroclaw University of Economics and Business, 53-345 Wroclaw, Poland

**Keywords:** goose meat, cooking, energy value, mineral concentration, chemical composition, retention coefficient

## Abstract

**Background:** Heat treatment methods including frying (with and without fat or oil), deep frying, oven roasting, grilling, charcoal roasting, broiling, steaming, and microwave cooking promote a cascade of adverse changes in the functional properties of meat, including protein fraction, lipid oxidation, and loss of some vitamins and mineral compounds. The aim of this study was to evaluate the influence of three cooking methods (sous vide (SV), microwave (M) cooking, and stewing (S)) on the basic chemical composition, cholesterol content, energy value, mineral concentration, and retention coefficients in goose meat. **Methods:** Basic chemical composition and mineral analysis were determined using AOAC methods. Total cholesterol content was established using the HPLC method. **Results:** Both types of goose meat (without and with skin) and heat treatment had a significant effect on nutrient values, mineral concentration, and retention coefficients. The S meat was characterized by a higher protein content than M and SV meat, and had the lowest fat, protein, and cholesterol retention, among other methods. The M meat had lower total cholesterol content than SV and S meat. There were significant differences in energy value for SV, M, and S meats. The SV meat contained less P, Mg, Fe, Zn, and more Na and K than the M and S samples. The highest values of Zn, Mg, and Fe content and the lowest of K and Ca were recorded in S meat compared with the SV and M samples. The retention coefficients of P, Mg, Na, Ca, and K in S meat were lower than in the SV and M samples. The meat without skin was characterized by a lower energy value, fat content, retention of proteins, and cholesterol, but higher fat retention than skin samples. This meat contained more minerals such as P, Mg, Fe, K, Na, and less Ca than skin meat. Higher retention coefficients were observed for Zn, P, Mg, Ca, and lower were observed for Na, Fe, and K in meat without skin than in samples with skin. **Conclusions:** From a dietary point of view, the most beneficial were SV muscles without skin. Whereas, taking into account the protein, fat content, and retention coefficients of fat, cholesterol, Zn, and Na, the most optimal form of cooking for meat with skin seems to be stewing. These results may be used by consumers in making dietary choices by taking into account the type of goose meat and kind of heat treatment.

## 1. Introduction

Consumers are encouraged to avoid highly processed meat products, especially products heated to a high temperature. During heat treatment, adverse changes occur in the functional properties of meat, including protein fraction, lipid oxidation, and loss of some vitamins and mineral compounds. Moreover, during some thermal processes (for example microwave cooking, pan frying, or grilling) the meat may undergo the formation of heterocyclic aromatic amines, polycyclic aromatic hydrocarbons, and other substances harmful to humans [1,2,3,4,5,6,7]. Therefore, people’s preferences have changed to minimally processed, easy-to-eat foods with the least additives. The industrial sector has noticed changes in consumer demand; thus, producers have focused on accelerated food preparation and presentation techniques, half-ready and ready-to-eat foods [8]. The sous vide technique is a product alternative that fulfills consumer demands for ready-to-eat foods [9]. The early 2010 s saw a massive increase in sous vide cooking in restaurants and households. The word sous vide originates from French, meaning “under vacuum”, and is defined as cooking food in heat-stable vacuumed containers under a controlled temperature for a specific time followed by low-temperature storage [10,11]. Sous vide cooking generally uses low temperatures (50–85 °C) for a more extended time, depending on the type of meat [2,12]. After heating, the products are rapidly cooled to temperatures around 0–3 °C [13]. Sous vide applies to almost all types of foods. However, the recommended heat treatment parameters in sous vide dark poultry meat processing oscillate in the range of 70–80 °C. For example, duck meat is usually cooked at 80 °C until it is fall-apart tender, about 4–6 h and 8–12 h at 70 °C [10].

The most popular types of consumed meat are pork, beef, lamb, turkey, and chicken, while the meat of waterfowl poultry, including goose meat, is consumed in much smaller amounts. Despite its specific nutritional and culinary value, due to its high price, it is also less popular than gallinaceous poultry meat [14,15]. In Poland, the primary breed used to produce goose meat is White Kołuda^®^ geese, also known as “Polish oat geese” because the birds are fattened freely with oats in the last three weeks of rearing. Fattening with oat gives the goose meat and fat unique health-promoting and taste qualities [14]. The alternative to the most popular heat treatment methods for goose meat may be sous vide cooking. This method generates improved tenderness, juiciness, color, and flavor, and a reduction in protein damage, deterioration of lipids, and other heat-sensitive compounds [16,17]. To minimize nutrient loss while optimizing the palatability and shelf life of meat during thermal treatment, many food processors now use the sous vide cooking technique instead of conventional methods [18].

The sous vide method is a recent, popular heat treatment method used for beef, veal, lamb, pork, and gallinaceous poultry meat [3,13,19,20,21,22,23]. There is a great gap in the scientific literature that highlights goose meat quality while comparing conventional and sous vide techniques. Therefore, our study focused on analyzing basic chemical and mineral concentrations, cholesterol content, calculation of the energy value, and determination of nutrient retention coefficients of goose meat subjected to sous vide cooking, and comparing conventional methods such as stewing and microwave cooking. This work complements our previous research [5,6,7] on the effect of various heat treatment methods on the quality of goose meat.

## 2. Materials and Methods

### 2.1. Meat Samples

Material for the study consisted of breast (*Pectoralis major*) muscles obtained from the carcasses of 17-week-old females of White Koluda geese named “Polish oat geese”. The geese are fed and maintained in a specific way. The geese are kept in open-air runs and at pasture. They are reared and fattened up to 17 wks of age according to the standard, Polish fattening technology of White Kołuda geese. From the 14th to 17th wks of age, the birds are fattened freely with oats, which is why they are named ”Polish oat geese.” The birds in this study were fed a complete concentrated diet [5]. The birds were slaughtered in a commercial slaughterhouse in Poland respecting EU regulations (Council Regulation EC No 1099/2009). The eviscerated carcasses were placed into a 2 °C to 4 °C cooler for 24 h and then the breast muscles were cut out. The breast muscles were standardized for thickness and weight (average weight for breast muscles with skin and subcutaneous fat 485.5 ± 48 g, without skin 389.7 ± 35 g). The 48 breast muscles (24 with skin and subcutaneous fat and 24 without skin) were investigated.

### 2.2. Heat Treatments

Sous vide (SV), stewing (S), and microwave (M) cooking methods were tested. No food additives were used in the trials. Twelve geese breast muscles (6 samples with skin and 6 samples without skin) were used in each heat treatment. The final cooking temperature (75 °C) in all of the heat treatments was monitored in the geometric center of each muscle by inserting a Teflon-coated thermocouple (Type T, Omega Engineering Inc., Stamford, CT, USA) connected to a temperature recorder (VAS Engineering Inc., San Diego, CA, USA).

After heat processing, muscles were allowed to cool to room temperature. Then, the muscles were stored at 4 °C for 24 h in the refrigerator. Next, they were allowed to equilibrate to room temperature (21 °C, 3 h). The basic chemical composition, cholesterol content, and mineral composition were determined. 

### 2.3. Sous Vide (SV)

The temperature of 70 °C and time of 4 h were used for the sous vide method. Before cooking, each breast muscle was weighed and placed into a polyamide/polyethylene bag with the following parameters: thickness of 92 μm, gas permeability (O_2_, N_2_, and CO_2_) of up to 50 cm^3^/m^2^, and water vapor permeability up to 4 g/m^2^. These bags can be used at temperatures from −40 °C to +120 °C and are suitable for vacuum packaging machines. The samples were packed in a vacuum sealing machine (Profi Line 40+, Hendi, Robakowo, Poland) with a vacuum of 99.6%. After that, the samples were submerged in a thermostatic water bath (model SW 22, Julabo GmbH, Seelbach, Germany) preheated to 70 °C. The heating time of 4 h was applied once the core temperature of the muscles reached the water bath temperature of 70 °C (a hand-held thermometer was used, thermometer, DT-34 with a probe, Termoprodukt, Bielawa, Poland). Once the cooking process was finished, the pouches were removed from the water bath and submerged in ice-cold water (2 °C) for 1 h. Subsequently, the packed breast muscles were kept under refrigeration (4 °C for 24 h).

### 2.4. Stewing (S)

Each breast muscle was first pan seared with goose lard (10 g) for 1 min per side and then stewed. The muscles were individually placed in a stainless steel pan, and 250 mL of hot water (100 °C) was added (the muscles were covered with water). Next, the samples were covered with a lid and simmered for approximately 1 h and 30 min.

### 2.5. Microwave Cooking (M)

For microwave cooking, the samples were separately placed on a ceramic container in a Pyrex^®^ pan in the center of the carousel of a Whirlpool microwave oven (power consumption: 900W-Mod. MWP 253 SX, Whirlpool EMEA, Wroclaw, Poland). Two heating cycles of 4 min (700 W power setting) were used, and the muscles were turned over between processes (internal temperature 75 °C).

### 2.6. Basic Chemical Composition and Energy Value

Composition properties (protein, fat, moisture, and ash content) of goose raw and heat treatment meat with and without skin were determined using AOAC methods [24]. Before analysis, the breast muscles were ground separately (mesh size 3 mm) and completely homogenized with an IKA homogenizer (SBS-MR-2500, IKA-Werke GmbH & Co., Staufen, Germany). The moisture (%) content was calculated by weight loss after 12 h of oven-drying of the samples (3 g) at 102 °C (to constant weight) in a Memmer laboratory dryer (UN 75, Schwabach, Germany) (950.46B, p. 39.1.02). Crude protein (%) content was determined by the Kjeldahl method with an automatic Kjeldahl nitrogen analyzer (Kjeltec 2300 Foss Tecator distiller Häganäs, Sweden) (992.15, p. 39.1.16). For conversion of nitrogen into crude protein, a factor of 6.25 was used. Fat (%) content was measured by the Soxhlet method with petroleum ether extraction using a Hanon Automatic Soxhlet Extractor (SOX 606, Hanon Advanced Technology Group Co., Ltd., Jinan, China) (960.39 (a), p. 39.1.05). Ash (total mineral content in %) was determined by incineration at 550 °C for 10 h in an FCE 7SHM muffle furnace (Czylok, Jastrzębie Zdrój, Poland) (920.153, p.39.1.09). Energy Value was derived by multiplying the amount of protein and fat by conversion factors 4 and 9, respectively [25].

### 2.7. Total Cholesterol Content

Preliminarily ground breast muscles were homogenized in a T 25 homogenizer (Ika Ultra-Turrax Corp., Staufen, Germany). Lipid extraction for cholesterol content determination was carried out according to the procedure described by [26]. To quantify cholesterol, hot saponification was performed according to [16]. The preparation of trisilyl cholesterol derivatives for chromatographic analysis was performed based on the procedure described by Cunha et al. [27]. The cholesterol was quantified by the gas chromatography method using an HP-5 capillary column (30 m × 0.32 mm × 0.25 µm) (Agilent Technologies Inc., Santa Clara, CA, USA) and a flame-ionization detector (FID) in gas chromatograph 7820 A series (Agilent Technologies Inc., Santa Clara, CA, USA). The carrier gas (nitrogen) flow rate was 3 mL/min. The cholesterol peaks were identified by comparing retention times with the standard (5α-cholestane). The cholesterol content was expressed as mg/100 g of wet samples [28].

### 2.8. Mineral Analysis

The freeze-dried (temperature at −55 °C) ground goose meat (0.5 g for Ca, Mg, K, Na, and 1.0 g for Zn and Fe) was wet digested with 2 mL of H_2_O_2_ (30%) and 8 mL HNO_3_ (65%) using a microwave oven (Microwave Speedwave Digestion EXPERT, Berghof Products and Instruments, Engen, Germany). Digestion was conducted by applying a 4-step program. Digested samples were cooled, placed in polypropylene tubes, and diluted to 20 mL with ultra-pure water. A blank digest was made in the same way. Four macroelements (K, Na, Ca, and Mg) and two microelements (Zn and Fe) were determined via flame atomic absorption spectrometry using a spectrometer (model VARIAN AA 140, Varian, Mulgrave, Australia) according to the procedures of AOAC [29]. The mineral content in the samples was expressed in milligrams per 100 g of dry mass (DM). The phosphorus content was evaluated after previous mineralization of samples with HNO_3_ (65%) and HClO_4_ acid in a close microwave mineralizer (model MARS 6, Microwave Digestion System CEM corporation Matthews, USA). It was analyzed spectrophotometrically by the ammonium vanadomolybdate method using a spectrophotometer UV-VIS Shimadzu (model Shimadzu UV-1800, Osaka, Japan) at a wavelength of 470 nm [29]. The phosphorus content was expressed in milligrams per 100 g of DM.

### 2.9. Determination of Retention Coefficients

The percentage of nutrient retention after heat treatment was calculated using the following equation [30]:% Retention = (Nutrient content/100 g of meat after heat treatment/Nutrient content in 100 g of raw meat) × (meat weight (g) after heat treatment/meat weight (g) before heat treatment) × 100

### 2.10. Statistical Analysis

The results were log transformed to attain or approach a normal distribution and, subsequently, both a one and two-way analysis of variance (ANOVA) were employed in the orthogonal system. The statistical significance of the differences between the groups’ averages was verified using Tukey’s test, at the level of significance *p* ≤ 0.05, using Statistica^®^13.3 software [31]. The average values and their standard deviations are presented in the tables.

## 3. Results

### 3.1. Basic Chemical Composition and Energy Value

Table 1 shows the proximate composition of the “Polish oat goose” breast muscles raw and after undergoing three cooking procedures (microwave cooking (M), sous vide cooking (SV), and stewing (S)). In our study (for investigating moisture, fat, protein, and ash content), a significant (*p* = 0.001) interaction between meat type and heat treatment was found. Moisture content significantly varied from 73.1% in raw skinless to 56.9% in S with skin muscles (*p* ≤ 0.05). Moisture content showed wide variability between treatments. Compared with the raw goose meat, cooking led to a significant loss (by 3.5 to 19.7% calculated based on data provided in Table 1) of moisture for both kinds of meat. The muscles without skin were characterized by a higher content of moisture (62.9%) than those with skin (58.9%) (*p* = 0.001); this was probably due to greater moisture loss in cooking from the muscles with skin. The lowest content of moisture was observed for S (57.8%) and the highest for SV samples (60.0%) (*p* = 0.001).

The water loss in SV, M, and S samples results from structural changes in cooked meat. During cooking, the water is removed following pressure applied by the shrinking connective tissue on the aqueous solution in the extracellular void [32]. The composition and content of intramuscular fat are essential factors affecting meat quality and nutritional value. Fat is the most variable of the macronutrients in meat. Due to the thickening of the tissue structure after water loss, the fat content in the meat increases after heat treatment [33]. The heat treatment conducted resulted in significant changes in the fat percentage. Generally, the meat with skin was characterized by a higher fat content than skinless samples (13.4% vs. 5.3%) (*p* = 0.001). Raw muscles without skin contained less fat than that subjected to heat treatment. On the other hand, raw muscles with skin had more fat than cooked samples (for all investigated types of heat treatment). This probably resulted from the liquid phase formed during the cooking of the meat, especially where meat with skin contains melted fat from subcutaneous fat. The decrease in moisture in skinless cooked muscle led to increased muscle fat, with no significant differences in the type of heat treatment (*p* ≤ 0.05). The cooked samples with skin subjected to stewing were characterized by the lowest fat content among the others (9.2% vs. 12.6%) (*p* ≤ 0.05).

Both kinds of raw muscles were characterized by lower protein percentages than cooked meat (Table 1). The breast muscles without skin showed a higher protein percentage (29.2%) than skinny samples (26.3%) (*p* = 0.001). The protein content in cooked samples increased by 30.9% to 94.9% compared with raw meat (calculated based on data provided in Table 1). The increased protein content in heat-treated meat resulted from moisture loss through cooking. The S samples were characterized by the highest protein content (33.9%) and SV meat had the lowest (27.9%). The protein content of both cooked meat samples was as follows: S > M > SV (*p* = 0.001). The protein structure modification (denaturation of sarcoplasmic and myofibrillar protein; heat shrinkage of the basement membrane collagen, endo-, peri-, and epimysial collagen; and complete gelatinization of epimysial followed by the peri- and endomysial collagen) and molecular changes (damage of amino acids residues; oxidative modification and protein carbonyls; aromatic residues; disulfide bond; Maillard reaction products; and food-derived heterocyclic amines) in meat protein are dependent on the cooking methods [34].

The ash content is an indicator of total mineral content in food. Ash is composed of macro- and microelements that are necessary for many changes in the human body. In our study, the heat treatment caused a significant increase in ash content compared with raw meat. Microwaved samples without skin were characterized by the highest (1.6%) ash percentage and S the lowest (1.4%) (*p* ≤ 0.05). In total, there were significant differences in ash percentage for meat between M (1.4%), SV (1.4%), and S (1.3%) cooking methods (*p* = 0.001). The ash content in cooked meat without skin was higher than in samples with skin (*p* = 0.001).

Generally, cooking results in the concentration of proteins, fat, and ash in cooked meat. Most nutrients increase their concentration due to moisture loss through cooking [35]. Such differences were caused by the fact that the liquid meat phase formed during the thermal process contains sarcoplasmic proteins and soluble minerals, and melts subcutaneous fat. In our research, all three cooking methods induced a comparable and statistically significant decrease in meat moisture. It was matched by a significant increase in the protein and lipid content and a consequent rise in the energy value. The energy value of raw goose breast muscles without skin (0.54 MJ/100 g) was lower than that of heat-treated muscles (0.69–0.79 MJ/100 g) (*p* ≤ 0.05). On the other hand, it was stated for raw samples with skin that the higher (1.01 MJ/100 g) energy value compared with cooked meat (0.90–0.95 MJ/100 g) because they had a significantly higher fat percentage. The heat processing increased skinless breast muscles’ energy value, which can be classified as follows: S > M > SV. The energy value of these cooked samples was higher by 46.3%, 38.8%, and 27.7% (respectively for S, M, SV—calculated based on data provided in Table 1) in relation to raw meat (*p* ≤ 0.05). The breast muscles without skin are characterized by a lower energy value than samples with skin (0.69 vs. 0.94 MJ/100 g) (*p* = 0.001) which is related to a lower fat percentage (5.3% vs. 13.4%). As expected, the energy levels of samples corresponded to fat content, and thus parts containing skin/fat had higher energy levels than those without. In cooked meat with skin, the M samples had higher energy values than the SV and S samples; there were no significant differences in energy value (*p* ≤ 0.05) between SV and S cooking methods.

### 3.2. Total Cholesterol Content

Both the type of goose meat (*p* = 0.001) and the kind of heat treatment (*p* = 0.001) influenced the cholesterol content (Table 1). All cooked samples had more cholesterol content than raw meat (*p* ≤ 0.05). Heat processing caused an increase in the cholesterol percentage in meat, ranging from 84.08 mg/100 g to 101.33 mg/100 g (depending on the type of meat and cooking methods) (*p* ≤ 0.05). It increased in relation to raw meat by 31.0 to 73.4% (calculated based on Table 1). The highest cholesterol content was stated for S meat without skin and the lowest for SV samples with skin (*p* ≤ 0.05). The breast muscles with skin were characterized by a lower (79.79 mg/100 g meat) cholesterol content than meat without skin (89.90 mg/100 g meat) (*p* = 0.001). There were significant differences (*p* ≤ 0.05) in cholesterol content in both types of meat for three kinds of heat treatment.

### 3.3. Mineral Concentration

Minerals are one of the classes of essential nutrients in the human diet. Deficiency or excess of certain essential nutrients is a concern for human health. As suggested by the present experiment, the cooking method may strongly influence the mineral content of meat.

Table 2 presents the outcome of macro- and microelements in goose meat. In our study investigating macro- and microelements, a significant (*p* = 0.001) interaction between type of meat and heat treatment was found. Cooked meat showed higher P, Mg, and Zn content than raw samples, except for K and Na in both types of meat and Ca in S meat with skin (*p* ≤ 0.05). In addition, heat treatment affected the various minerals in different ways. The highest content of Na and K characterized SV muscles without skin (185.09 mg/100 g and 1169.78 mg/100 g DM, respectively), and the lowest content characterized S meat with skin (147.10 mg/100 g and 710.86 mg/100 g DM, respectively), but there were no significant differences between the M and S samples (*p* ≤ 0.05). The meat without skin had more Na and K than samples with skin (Table 2). The K and Na minerals are present in meat in water-soluble form. The differences between the cooking techniques could be caused by the different mechanisms of water loss from the meat during the thermal process. The water loss in microwaving was caused mainly by water evaporation, which induces a concentration of minerals in cooked meat tissue. In sous vide cooking, evaporation was limited by plastic pouches, but the minerals were lost along with thermal leakage. In stewing, the meat is immersed in the water environment and, probably, the minerals are leached into the cooking water. Regarding Ca content, the highest value was noted for M samples with skin (45.83 mg/100 g DM) and the lowest for S-meat without skin (22.80 mg/100 g DM); it was increased by 109.9% and 9.1% (calculated based on data provided in Table 2) compared with raw meat, respectively. As for S meat with skin, only the Ca content was lower (by 3.7% calculated based on data provided in Table 2) than in raw samples (*p* ≤ 0.05). The calcium content in meat with skin (33.20 mg/100 g DM) was higher than in skinless meat (28.81 mg/100 g DM) samples (*p* = 0.001). The iron content in cooked meat ranged from 17.41 mg/100 g DM in SV meat with skin to 24.94 mg/100 g DM in S samples without skin. Significant lower changes in relation to raw meat were recorded in Fe content for SV than M and S meat (*p* = 0.001) (Table 2).

The highest increase (by 70.7%—calculated based on data provided in Table 2) in iron content compared with raw meat was stated for S meat (*p* = 0.001). From a health point of view, it is worth noting that none of the applied types of heat treatment decreased the iron content in the meat. This was probably caused by heat denaturation of the myoglobin molecule and the increase in insoluble heme iron, and hence a change in the heme to non-heme iron ratio. On the other hand, non-heme iron is dangerous for lipid oxidation in meat. Iron has an essential role in human health. Its deficiency in the human body can lead to anemia and impediments to child growth and development. Iron is present in the form of heme (hemoglobin and muscle myoglobin), accounting for its high bioavailability in meats [35].

The mean values of Zn, P, and Mg in both kinds of cooked samples increased significantly compared with raw meat. Zn content in cooked meat ranged from 7.30 to 9.20 mg/100 g DM and was significantly higher for S in relation to the M and SV samples (*p* = 0.025). The highest increase (by 157.3% calculated based on data provided in Table 2) in Zn concentration was related to the S samples without skin (*p* ≤ 0.05). There were no significant differences in Zn content between meat without and with skin (*p* = 0.001). The highest values of P and Mg in cooked goose meat were established for M samples without skin (1142.38 mg/100 g and 116.19 mg/100 g DM, respectively) and the lowest value was for SV meat with skin (950.48 mg/100 g and 96.80 mg/100 g DM, respectively) (*p* ≤ 0.05). The meat without skin contained more P and Mg (964.88 mg/100 g and 102.29 mg/100 g DM, respectively) than that with skin (897.66 mg/100 g and 99.15 mg/100 g DM, respectively) (*p* = 0.001). The highest increase in P content in relation to raw meat was observed for S meat with skin (by 83.3% calculated based on Table 2) and Mg concentration for M meat without skin (by 62.2%).

### 3.4. Retention Coefficients

Additionally, the retention values were calculated to evaluate the increase in or loss of nutritional compounds during thermal treatment. The liquid phase, which is formed during the cooking of the meat, mainly contains sarcoplasmic proteins, minerals, vitamins, and melted fat. The energy value of the meat depends on the extent to which these basic meat components are retained in the meat subjected to the heating process. Retention of nutrients in food subjected to cooking is vital from a dietary point of view [6]. In our research on basic meat components, the retention in breast goose muscles ranged from 23.4% (fat retention in S meat with skin) to 109.0% (protein retention in M samples with skin). It depended on the type of meat and the heating process. In the case of protein, retention of around 100% and more is expected, deriving from the sheer effect of concentration. The higher protein retention characterized SV and M than S meat; the samples with skin had higher retention than those without skin (*p* = 0.001). In the case of fat retention, the meat without skin showed a higher value (82.2%) than those with skin (36.0%). Such a low-fat retention rate in meat with skin could have been due to the significant reduction in fat content (by 35.4–52.9% calculated based on data in Table 3) in muscles. This variability has been explained by unpredictable levels of subcutaneous and intermuscular fats, which liquefy during cooking. In addition, S samples were characterized by the lowest (47.2%) fat retention compared with SV and M samples (*p* = 0.001). The dripping and leaching losses of minerals have an important effect on ash retention in cooked meat [30] and such an effect on ash retention was found in the present study. No effect of the type of meat and interaction between the kind of meat x heat treatment on ash retention was observed (*p* = 0.074; *p* = 0.058, respectively). However, a significant impact of heat processing was found (*p* = 0.001). The significantly lower (67.2%) ash retention was stated for S samples in comparison with SV (93.6%) and M (90.0%) muscles (*p* = 0.001). This could be because stewing is a wet method, as opposed to the microwave (a dry method), and the SV method, where a plastic bag and vacuum prevent the loss of minerals. A lower cholesterol retention was found in meat without skin (93.3%) than with skin (99.2%) (*p* = 0.001). The apparent retention of cholesterol was around 100% and a little higher than 100%, suggesting it had been concentrated with migration from the subcutaneous and intermuscular fat. There was an interaction between the kind of meat x heat treatment (*p* = 0.001). The SV samples had the highest (106.0%) value of cholesterol retention and S meat had the lowest (86.2%) (*p* = 0.001). The differences in cholesterol retention coefficients between samples without and with skin in the case of SV and S meat are not stated.

Regarding minerals, a significant (*p* = 0.001) influence of the type of meat and cooking methods on retention of Zn, Fe, Na, K, Ca, and Mg was observed. No interaction between kind of meat x type of heat treatment was stated for phosphorus retention (*p* = 0.097). Depending on the applied factors, mineral retention ranged from 25.2% of Na for S meat without skin (the lowest value) to 152.2% of Zn for M (the highest value). Regarding the type of heat treatment, the lowest retention of Na was stated for S meat (28.7%), while the highest retention of Zn was for M meat (142.7%) (Table 4). The lowest retention of P, Mg, Na, K, and Ca was found for S meat than for SV and M samples. During the stewing process, inorganic materials such as Na, K, Ca, Mg, and P present in food are easily lost with water. In the case of Zn retention, there were no significant differences between S (152.0%), M (152.1%), and SV (148.4%) meat. Zinc is, among others, the most stable mineral. The Zn, P, Mg, and Ca retentions in samples without skin were higher and Na, Fe, and K were lower than in skin meat.

In general, meat without skin is characterized by a lower energy value, fat content, retention of proteins, and cholesterol, but higher fat retention than skin samples. This meat contained more minerals such as P, Mg, Fe, K, Na, and less Ca than skin samples. Higher retention coefficients were observed for Zn, P, Mg, Ca, and lower retention was observed for Na Fe, and K in meat without skin than in samples with skin. In terms of the kind of heat treatment, there were no significant (*p* = 0.001) differences in energy value and fat content in the SV, M, and S samples. The S meat is characterized by a higher protein content than the M and SV meats, with the lowest fat, protein, and cholesterol retention, among other methods. The M meat had lower total cholesterol content than the SV and S meats. Regarding mineral concentrations, the SV meat contained less P, Mg, Fe, and Zn and more Na and K than the M and S samples. The highest Zn, Mg, and Fe content and the lowest K and Ca compared with the SV and M samples were stated in S meat. The retention coefficients of P, Mg, Na, Ca, and K in the S meat were lower than in the SV and M samples. The M and SV cooking in this study did not require water, which probably allowed for higher retention of the main nutrients and minerals than in the stewing process.

## 4. Discussion

### 4.1. Basic Chemical Composition and Energy Value

The data obtained in our experiment were in harmony with those stated by [36]. In Turkish goose breast meat, they observed that the cooking process (seven different heating procedures) caused an increase in the dry matter because of the reduction in the water content. In cooked meat, the water content ranged from 53.8 to 62.9%. The lowest moisture content was found in the microwaved (M) samples. In our study, the moisture for M “Polish oat goose” meat was similar (57.9%) (*p* = 0.001). They stated no significant differences in protein content between M, oven-roasted (OR), boiled (B), deep-fried (DF), pan-fried without (PFW), and with oil (PFO) samples; only grilled (G) meat had a lower percentage of protein. In our study, the protein content varied depending on the cooking procedure, but the differences in fat content were not significant. Based on a study by Abdul-Naeem et al. [37], the total volatile basic nitrogen (TVBN) is considered as a clear indicator of incipient protein deterioration due to accumulated ammonia, amine, and trimethylamine. Such higher TVBN values observed in the samples may be attributed to the higher protein content in these samples as compared with the other cooking methods. The study by [36] showed a higher fat content in G and M meat than in others. Goluch et al. [6] stated that protein, fat, and ash were concentrated due to the loss of water in the meat subjected to boiling, grilling (G), roasting (OR), and frying in a pan without oil (PFW). The application of various heat processing methods also significantly impacted the energy value of goose meat. The moisture contents in our experiment concerning SV (both kinds of meat) were slightly higher than in the G, OR, and PFW samples. The protein percentage in the S samples was similar to G, but a slightly lower protein content characterized M and SV meat than the remaining ones. The highest ash content was stated for G (1.7%) samples without skin and in our experiment for M ones (1.6%) (*p* ≤ 0.05). Our results (23.4–109.0%) concerning the retention of basic nutrients in M, SV, and S samples corroborate the previous findings (28.3–103.0%). Contrary to our findings, in the Goluch et al. [7] study, neither the skin presence nor the type of treatment influenced the protein content. The current experiment stated a higher protein retention for meat without skin than those with skin (*p* = 0.001). The lowest retention has been shown for the fat in muscles with skin, and these results were in harmony with data obtained by Goluch et al. [6]. In this experiment, the energy value of cooked goose meat increased by 25.5–32.9% compared with raw meat, depending on cooking methods. In the current study, the energy value of M, SV, and S samples increased only by 2.6 to 8.9%. Likewise, Belinsky and Kuhnlein [38] reported that heat processing (fire-roasting FR, B, and OR) of Canadian goose breast muscles with skin influenced moisture, protein, fat, ash content, and energy value in cooked meat. It was observed that protein content ranged from 28.50 to 34.00%, fat from 10.1 to 18.0%, moisture from 50.6 to 56.1%, ash from 0.7 to 0.9%, energy value from 0.93 to 1.2 MJ/100 g, and depended on the type of heat processing. The energy value was as follows: OR > B > FR, but the differences were insignificant. These values are slightly different from our results for the M, SV, and S samples with skin. In turn, Głuchowski et al. [39] observed that the applied cooking methods (boiling, steaming—ST, and SV at 75 °C) reduced the water content and increased the protein and fat content. The SV samples were characterized by significantly higher fat (1.94%) and lower protein (24.5%) content than boiled chicken meat (1.42% and 29.20%, respectively). Ramane et al. [40] showed that SV (at 80 °C) cooking of chicken skinless breast muscles reduced moisture content, but protein, fat, and ash contents were increased compared with raw meat. Karpińska-Tymoszczyk et al. [22] observed that water content significantly decreased and protein and fat percentage increased in SV-cooked (at six different combinations of temperature and time) chicken breast fillets, but the differences depended on the process conditions. Rasińska et al. [41] related that SV-cooked rabbit meat had higher moisture than the boiled or oven-roasted ones. They explained that the sous vide meat is packed in vacuum pouches and kept in a water bath at 50–85 °C, minimizing the damage to heat-sensitive proteins and nutritional compounds. Compared with raw rabbit meat, cooking led to a significant increase in lipid content. Macharáčková et al. [42] stated that moisture content in pork meat after SV cooking at the temperature of 70 °C for 3 h (SV 70 °C/3 h) was higher than those of raw meat by 16.6%, but there was no significant difference with broiled samples. A higher protein content was found in broiled meat compared with SV (70 °C/3 h), but there were no differences in fat contents. In contrast, the fat percentage was higher in raw meat, similar to our findings for goose meat with skin. In turn, Choi et al. [43] found that B and OR chicken meat produced significant (*p* < 0.05) lower moisture content, along with a significantly higher protein content as compared with M samples. All samples’ fat and ash contents were not significantly different (*p* > 0.05). Echarte et al. [44] reported that both M heating and PF significantly decreased the moisture of beef and chicken patties. Microwave heating did not increase the lipid percentage, and PF slightly increased it only in chicken. According to Abdel-Naeem et al. [37], B and PF rabbit meat resulted in significantly lower moisture content, along with significantly higher protein and fat contents in comparison with microwave cooking (*p* < 0.05). In the study of Alfaia et al. [45], it was stated that cooking led to a significant loss of moisture and, consequently, to a significantly higher fat content, with differences (*p* < 0.05) among M, B, and G beef meat. They reported a higher fat retention for both M (121.7%) and G (120.9%) beef meat when compared with B (106.6%) ones. According to Kirchner et al. [46], the microwaved beef patties tended to have less fat and more protein than broiled (BR), OR, and PF. The moisture content in beef steak depended on the cooking methods and was classified as PF > OR > M > BR samples. They established that M beef patties contained statistically less (0.71 MJ/100 g) energy (but with no practical significance) than other (BR, OR, and PF) cooked meat (0.77–0.79 MJ/100 g). Whereas the percentage of fat retention in beef patties differed and was as follows: PF (99.5%) > OR (96.8%) > BR (92.2%) > M (87.8%). In our study, the percentage of fat retention in M goose meat without skin was similar (86.7%) to that for M beef patties. Serrano et al. [47] stated that moisture, protein, and ash content in cooked beef steaks compared with raw meat was significantly higher, but the fat percentage was higher only in PF steaks. The M and OR meat had lower moisture and higher protein content than others (*p* < 0.05). The energy value of cooked beef meat ranged from 96.3 to 110 kJ, but the differences were insignificant compared with raw meat and between heating procedures (M, OR, G, and PF). They observed that since protein is not susceptible to migration but only concentration, in beef steak subjected to various cooking techniques (M, OR, G, and PF), the protein retentions were around 100–104%. The fat retentions were the lowest in M meat (62%) and the highest (153%) for PF steak. However, ash retention for cooked meat ranged from 73.0 to 88.0%, and the lowest was for M ones. The protein and fat retention coefficients agreed with our M and SV goose meat. Nudda et al. [48] performed M cooking of lamb meat. As expected, the cooking process reduced the meat moisture by 7.3%, with a consequent increase in fat (by 46.2%) and protein (by 23.4%) concentration, thus determining an increase in energy value (by 26.6%). In our experiment, the energy value for microwaved goose meat without skin increased by 38.8% but decreased by 5.9% for meat with skin. According to Campo et al. [49], the percentage of protein and fat in S lamb meat increased by 45.5% and 54.1% (respectively) compared with the raw meat, and it was higher than for G samples (23.9% and 26.0%, respectively). In our study, the protein content in S meat increased by 74.3% compared with the raw meat and the fat percentage in meat without skin by 21.5%. Maranesi et al. [30] established that M cooking provided higher (72.6–113.0% depending on nutrient) retention coefficients than broiling (BR) (72.4–101.0% depending on nutrients) in lamb meat. The lipid retention was 96% and 102% in BR and M cooking, ash 72.4% and 72.6% (respectively, but there were no significant differences), and protein 101% and 113% (respectively, and they differed significantly).

### 4.2. Total Cholesterol Content

Echarte et al. [44] observed a significant increase in cholesterol content in beef and chicken patties with M heating and PF. In M beef patties, the cholesterol content increased by 13.7%, and in PF by 16.3%. However, the content in chicken meat increased by 20.2% and 23.8%, respectively, compared with raw samples. Compared with data, in our research, the total increase in cholesterol content was much higher in M goose meat (44.9%), but lower than for SV and S samples (*p* = 0.001). Slover et al. [50] stated that S pork loin meat had more cholesterol content than OR meat. The cholesterol content in OR meat increased by 41.1% compared with raw meat, but in S samples it increased by 72.9%. They stated that fat and cholesterol retentions in S pork loin meat were 117% and 102%, and higher in OR meat (141% and 105%, respectively). On the other hand, Rodriguez-Estrada et al. [51] established that the amount of cholesterol present in the raw beef sample was significantly higher than in cooked (barbecued—BA, B, M, OR, and PF) samples, and the cholesterol content varied greatly. The total cholesterol content in B samples was significantly lower than in M meat. In comparison with raw beef, the cholesterol content in M meat decreased by 29.7%. Kirchner et al. [46] did not observe significant differences in cholesterol content in M (59.7 mg/100 g) beef patties and OR and PF samples (56.3–59.6 mg/100 g). However, the cholesterol content in meat can also decrease due to heat treatment as cholesterol is heat labile.

### 4.3. Mineral Concentration

According to the other author’s results [3,6,35,38,42,49,51,52] the mineral composition in cooked meat varied depending on meat type and the heat treatment method used. In a previous study conducted by Goluch et al. [6], the phosphorus contents in goose meat cooked with four methods (WBC, OR, G, and PFW) were lower than the results we obtained for the M, SV, and S samples. Contrary to the current results, the P content in cooked goose meat without skin found by Goluch et al. [6] was lower than in raw samples, but higher, similar to our findings, in samples with skin. Similar to the current study, Goluch et al. [6] stated higher P, Na, K, Mg, Fe, and Zn contents in goose muscles without skin than with skin. The content of Ca and Fe indicated in WBC muscles without skin was the highest and significantly higher than in raw meat. The highest value of Ca and Fe was noted in examined SV and M muscle samples with skin. Regarding Zn, they found that a significant impact of heat processing was only evident in muscles with skin. In contrast, heat processing in the current study increased Zn contents significantly in both kinds of meat. Goluch et al. [6] observed a significant influence of the muscle type and heat processing on the retention of all minerals. Generally, the lowest retention was determined for Na in OCR muscles without skin, and the highest was for Ca in WBC samples. We observed the lowest retention of Na and K in S muscles but the highest Zn was in M meat (with and without skin). We observed that the presence of skin in meat affects the Fe and Zn retention, while the treatment method only influenced the retention of Fe. In a study performed by Oz and Celik [36], no differences in Na, Ca, Mg, P, and Zn contents were established for seven applied cooking methods in Turkish goose meat. Heat processing significantly increased only the contents of K (except boiling) and Fe (all methods) compared with raw meat (P < 0.05). The concentrations of Fe, Ca, and Zn minerals in Canadian goose breast muscles was higher in cooked samples than in raw meat and is subject to the kind of heat treatment [38]. Depending on cooking methods, the contents of Fe, Ca, and Zn were as follows: B > FR > OR; OR > FR > B; B > OR > FR (respectively). In turn, in the study of Lopes et al. [35], cooked (B, M, and G) veal meat showed higher mineral (Fe, Mg, Zn Mg, and Ca) contents compared with the raw samples, except for K. They related that most of the minerals were reasonably well retained by cooking. The Fe and Zn retention values varied between 78.9–105% and 88.9–97.0% for M and G veal (respectively). However, Mg retention ranged from 93.4–138.0% and K between 26.6–68.4% for B and M meat (respectively). Moreover, Gerber et al. [52] reported an increase in the Fe, Zn, and Ca concentrations in grilled, boiled beef and decreased K and Na minerals in G meat compared with raw meat. According to Modzelewska-Kapituła et al. [3], all thermal processing methods caused an increase in Ca, Zn, and a decrease in K and Na contents compared with raw beef meat. Steam-cooked (ST) beef showed higher Ca, Fe, and Zn contents and lower contents of Na and K than the SV samples. The content of Mg in the S and SV samples was similar to that noted in raw meat. The mineral retention coefficients varied from 57.8% to 110.6% and the highest retention was noted for Zn in ST meat. Retention values above 100% were noted only for Cu and Zn in ST beef, whereas the remaining minerals (Ca, Fe, Mg, Mn, K, and Na) were below 100%, indicating loss of minerals during cooking. In this study, the thermal treatment method for retaining higher mineral contents in beef (including Cu, Zn, and Fe) was steam cooking [3]. Macharáčková et al. [42] established that the Na, K, and Fe concentrations decreased in SV and broiling (BR) pork meat compared with raw samples. The decrease in the content of these minerals and changes in Mg, Ca, and Zn contents significantly depended on the temperature and time of the SV procedure. The retention rates evidenced the gentle effect of the SV method. In this regard, the highest retention rates for Zn, Fe, Ca, Na, and Mg were achieved by SV at 55 °C. The lowest retentions were noted for SV at 60 °C/12 h for Zn, Cu, and Fe trace elements. The temperature duration had an effect on these elements. Analogous to our experiment, the major minerals present in meat as relatively simple inorganic compounds, such as K and Na, are more likely to be released. Nikmaram et al. [53] noted that the Na content in M, S, and OR veal meat decreased; they concluded that Na, which is water-soluble, would be incorporated into the cooking loss associated with soluble proteins. Indeed, this drop in Na was greater in S than in the other two heat treatments. It was observed that despite significant fluid loss in the M treatment, more Na was lost during the stewing treatment in which water was used as an auxiliary liquid for cooking. Similarly, in our study, the Na content for S meat without skin was lower than for M samples, but the difference was insignificant. Campo et al. [49] analyzed Fe and Zn content in lamb meat subjected to OR, G, and S. No significant differences were found among the cooking treatments in Fe content, and OR lamb meat showed higher Zn content than the S or G samples.

Our results agree with the other presented reports and validate that the concentrations of most nutrients increased due to cooking-induced moisture loss. This situation is seen in different kinds of meat. These changes in cooked meat depend on cooking methodology and are the result of mass transfer during thermal treatment.

Heat processing significantly impacted the concentration of iron in muscles. This could be related to larger water losses that take place during heat processing, and the high temperature leads to fast denaturation of myoglobin that can release iron from heme [6]. Moreover heme iron is converted after different heat treatments to non-heme iron, and this iron is the less available form of this mineral. Moreover, sodium, which is the major cation in extracellular fluids, can decrease with leakage. The loss of this nutrient along with thermal leakage during cooking is significantly higher than intracellular ions and minerals bound to proteins. During the heat treatment processes, inorganic materials such as phosphorus and calcium are extracted with cooking loss. There is general agreement that zinc, copper, and iron are the most stable minerals in cooked meats and the degree of meat shrinkage during cooking significantly affects the retention of minerals [6]. On the other hand, the greater retention of minerals in meat results from the specificity of processing. For example, microwave cooking and sous vide did not require water, which probably allowed for higher retention of the minerals. Moreover in high temperatures, proteins form a “shell” on the surface of the meat so the soluble minerals, such as nitrogen and inorganic salts, are lost to a lesser extent [54]. In addition, divalent minerals are better retained during processing than monovalent minerals, which may be due to their stronger relationship with proteins. Skin can provide a kind of protection (barrier) for the muscles against water loss during cooking, hot air during baking, and hot plates during grilling and oil frying [6,52].

Further research will be carried out on the determination of the optimum temperature and processing time for goose meat by the sous vide method, taking into account the nutritional value, functional properties of the meat, and the substances adverse to the health of the consumer arising in it during heat treatment.

## 5. Conclusions

It has been demonstrated that in both types of goose meat (with and without skin), heat treatment (M, SV, and S) significantly affected nutrient contents, mineral concentrations, and retention coefficients. From a dietary perspective, the most beneficial were SV muscles without skin because they are characterized by the lowest energy value and fat content, as well as the highest retention of Mg, Ca, and K. Whereas taking into account the protein, fat content, and retention coefficients of fat, cholesterol, Zn, and Na, the most optimal form of cooking for meat with skin seems to be stewing. These results may be used by consumers in making dietary choices by taking into account the type of goose meat and heat treatment. Therefore, in a future study, we will present results on the impact of identical heat techniques on the changes in fatty acid profile, health lipid indices, and functional properties of “Polish oat” goose meat.

## Figures and Tables

**Table 1 foods-12-00129-t001:** Basic chemical composition, energy value, and total cholesterol content in raw and cooked goose meat (n = 6 breast muscles with skin and n = 6 without skin for each kind of heat treatment).

Parameters	Meat	Raw Meat (R)	Heat Treatment			Total	*p*-Value (*p* ≤ 0.05)		
			Microwave (M) Cooking	Sous Vide (SV) Cooking	Stewing (S)		Meat (Mt)	Heat Treatment (T)	Mt x T
Energy value (MJ/100 g)	Without skin	^y^0.54^d^ ± 0.02	^y^0.75^b^ ± 0.02	^y^0.69^c^ ± 0.03	^y^0.79^a^ ± 0.02	^y^0.69 ± 0.10	0.001	0.001	0.001
	With skin	^x^1.01^a^ ± 0.02	^x^0.95^b^ ± 0.03	^x^0.91^c^ ± 0.03	^x^0.90^c^ ± 0.02	^x^0.94 ± 0.05			
	Total	0.78 ± 0.24	0.85 ± 0.10	0.80 ± 0.12	0.85 ± 0.09				
Protein (%)	Without skin	^x^21.97^d^ ± 0.56	^x^31.35^b^ ± 0.47	^x^28.78^c^ ± 0.69	^x^34.81^a^ ± 0.53	^x^29.23 ± 4.81	0.001	0.001	0.001
	With skin	^y^16.94^d^ ± 0.40	^y^28.29^b^ ± 0.76	^y^26.99^c^ ± 0.59	^y^33.03^a^ ± 0.47	^y^26.31 ± 5.97			
	Total	19.46^d^ ± 2.64	29.82^b^ ± 1.69	27.89^c^ ±1.11	33.92^a^ ±1.04				
Cholesterol (mg/100g)	Without skin	^x^68.58^d^ ± 0.58	^x^89.89^c^ ± 0.53	^x^98.80^b^ ± 0.80	^x^101.33^a^ ± 0.63	^x^89.90 ± 13.11	0.001	0.001	0.001
	With skin	^y^54.08^d^ ± 0.96	^y^87.24^b^ ± 0.62	^y^84.08^c^ ±0.45	^y^93.77^a^ ± 0.62	^y^79.79 ± 15.51			
	Total	61.32^c^ ± 7.53	88.87^b^ ± 1.48	91.44^ab^ ± 6.63	97.55^a^ ± 3.95				
Ash (%)	Without skin	^x^1.21^d^ ± 0.05	^x^1.64^a^ ± 0.29	^x^1.58^b^ ± 0.18	1.38^c^ ±0.25	^x^1.45 ± 0.27	0.001	0.001	0.001
	With skin	^y^0.88^b^ ± 0.04	^y^1.20^a^ ± 0.02	^y^1.18^a^ ± 0.05	1.20^a^ ± 0.11	^y^1.12 ± 0.18			
	Total	1.05^b^ ± 0.18	1.42^a^ ± 0.32	1.38^a^ ± 0.27	1.29^a^ ± 0.20				
Water content (%)	Without skin	^x^73.08^a^ ± 0.62	^x^59.32^c^ ± 0.20	^x^60.53^b^ ± 0.23	^x^58.69^d^ ± 0.48	^x^62.91 ± 6.02	0.001	0.001	0.001
	With skin	^y^61.55^a^ ± 0.32	^y^57.91^c^ ± 0.62	^y^59.41^b^ ± 0.44	^y^56.88^d^ ± 0.67	^y^58.94 ± 1.85			
	Total	67.31^a^ ± 5.98	58.62^c^ ± 0.85	59.97^b^ ± 0.67	57.78^d^ ± 1.09				
Fat (%)	Without skin	^y^4.46^b^ ± 0.40	^y^5.76^a^ ± 0.60	^y^5.40^a^ ± 0.80	^y^5.42^a^ ± 0.50	^y^5.26 ± 0.75	0.001	0.001	0.001
	With skin	^x^19.50^a^ ± 0.41	^x^12.59^b^ ±0.74	^x^12.14^b^ ± 0.93	^x^9.19^c^ ± 0.62	^x^13.36 ±3.90			
	Total	11.98^a^ ± 7.77	9.18^ab^ ± 3.59	8.77^ab^ ± 3.58	7.31^b^ ± 2.02				

a–d: different letters in rows means statistically significant differences between group average, including thermal treatment (*p* ≤ 0.05); x–y: different letters in columns means statistically significant differences between group average, including kind of meat (*p* ≤ 0.05).

**Table 2 foods-12-00129-t002:** Mineral concentration in raw and cooked goose meat (mg/100 g DM) (n = 6 breast muscles with skin and n = 6 without skin for each kind of heat treatment).

Parameters	Meat	Raw Meat (R)	Heat Treatment			Total	*p*-Value (*p* ≤ 0.05)		
			Microwave (M) Cooking	Sous Vide (SV) Cooking	Stewing (S)		Meat (Mt)	Heat Treatment (T)	Mt x T
Zinc (Zn)	Without skin	^y^3.56^d^ ± 0.11	8.07^b^ ± 0.30	^y^7.19^c^ ± 0.19	9.25^a^ ± 0.24	7.02 ± 2.18	0.001	0.025	0.001
	With skin	^x^3.85^d^ ± 0.12	7.87^b^ ± 0.14	^x^7.41^c^ ± 0.18	9.15^a^ ± 0.22	7.07 ± 2.00			
	Total	3.70^d^ ± 0.19	7.97^b^ ± 0.25	7.30^c^ ± 0.21	9.20^a^ ± 0.23				
Phosphorus (P)	Without skin	^x^639.50^c^ ± 10.16	^x^1142.38^a^ ± 14.28	^x^1041.75^b^ ± 12.42	1035.88^b^ ± 17.44	^x^964.88 ± 196.12	0.001	0.001	0.001
	With skin	^y^568.17^c^ ± 4.83	^y^1030.43^a^ ± 19.04	^y^950.48^b^ ± 11.50	1041.58^a^ ± 6.01	^y^897.66 ± 196.86			
	Total	603.80^d^ ± 37.58	1086.41^a^ ± 60.05	996.12^c^ ± 48.53	1038.73^b^ ± 12.94				
Magnesium (Mg)	Without skin	^y^71.67^c^ ± 0.88	^x^116.19^a^ ± 1.63	^x^106.67^b^ ± 1.70	114.64^a^ ± 2.14	^x^102.29 ± 18.40	0.001	0.001	0.001
	With skin	^x^76.53^d^ ± 1.59	^y^108.79^b^ ± 1.37	^y^96.80^c^ ± 1.15	114.50^a^ ± 1.93	^y^99.15 ± 14.85			
	Total	74.10^d^ ± 2.80	112.49^b^ ± 4.08	101.74^c^ ± 5.29	114.57^a^ ± 1.97				
Sodium (Na)	Without skin	^x^364.66^a^ ± 2.11	^x^160.01^c^ ± 2.68	^x^185.09^b^ ± 2.04	^y^157.19^c^ ± 4.08	^x^216.74 ± 87.51	0.001	0.001	0.001
	With skin	^y^257.70^a^ ± 1.66	^y^154.23^d^ ± 2.99	^y^175.82^b^ ± 3.27	^x^147.10^c^ ± 2.44	^y^183.71 ± 41.29			
	Total	311.18^a^ ± 55.27	157.12^c^ ± 4.06	180.46^b^ ± 5.46	152.15^c^ ± 6.05				
Calcium (Ca)	Without skin	^y^20.89^d^ ± 0.44	^y^43.86^a^ ± 0.69	^y^27.67^b^ ± 0.70	^y^22.80^c^ ± 0.33	^y^28.81 ± 9.20	0.001	0.001	0.001
	With skin	^x^27.13^c^ ± 0.68	^x^45.83^a^ ± 0.87	^x^33.71^b^ ± 0.28	^x^26.12^d^ ± 0.45	^x^33.20 ± 8.00			
	Total	24.01^c^ ± 3.27	44.85^a^ ± 1.27	30.69^b^ ± 3.16	24.46^c^ ± 1.75				
Iron (Fe)	Without skin	^x^16.41^d^ ± 0.15	^x^24.18^b^ ± 0.29	^x^18.09^c^ ± 0.26	^x^24.94^a^ ± 0.33	^x^20.91 ± 3.78	0.001	0.001	0.001
	With skin	^y^12.24^d^ ± 0.33	^y^21.32^b^ ± 0.33	^y^17.41^c^ ± 0.20	^y^23.89^a^ ± 0.21	^y^18.72 ± 4.47			
	Total	14.32^d^ ± 2.16	22.75^b^ ± 1.51	17.75^c^ ± 0.42	24.42^a^ ± 0.61				
Potassium (K)	Without skin	^x^1472.63^a^ ± 7.75	^x^1057.41^c^ ± 9.47	^x^1169.78^b^ ± 10.34	^x^964.51^d^ ± 17.20	^x^1166.08 ± 194.71	0.001	0.001	0.001
	With skin	^y^825.30^a^ ± 14.00	^y^739.56^b^ ± 13.16	^y^750.39^b^ ± 5.23	^y^710.86^c^ ± 7.45	^y^756.53 ± 44.10			
	Total	1148.97^a^ ± 334.46	898.49^bc^ ± 164.49	960.09^b^ ± 216.72	837.69^c^ ± 131.61				

a–d: different letters in rows means statistically significant differences between group average, including thermal treatment (*p* ≤ 0.05); x–y: different letters in columns means statistically significant differences between group average, including kind of meat (*p* ≤ 0.05).

**Table 3 foods-12-00129-t003:** Nutrient retention coefficients (%) in raw and cooked goose meat (n = 6 breast muscles with skin and n = 6 without skin for each kind of heat treatment).

Parameters	Meat	Heat Treatment			Total	*p*-Value (*p* ≤ 0.05)		
		Microwave (M) Cooking	Sous Vide (SV) Cooking	Stewing (S)		Meat (Mt)	Heat Treatment (T)	Mt x T
Protein retention	Without skin	^y^95.68^a^ ± 0.54	^y^96.26^a^ ± 3.14	^y^92.55^b^ ± 2.38	^y^94.83 ± 2.75	0.001	0.001	0.001
	With skin	^x^109.01^a^ ± 2.66	^x^108.93^a^ ± 5.50	^x^96.83^b^ ± 2.03	^x^104.92 ± 6.84			
	Total	102.35^a^ ± 7.13	102.60^a^ ± 7.85	94.69^b^ ± 3.07				
Fat retention	Without skin	^y^86.66^b^ ± 1.61	^x^88.98^a^ ± 1.41	^x^70.92^c^ ± 0.88	^x^82.19 ± 8.29	0.001	0.001	0.001
	With skin	^x^42.15^a^ ± 0.89	^y^42.50^a^ ± 0.70	^y^23.41^b^ ± 0.64	^y^36.02 ± 9.14			
	Total	64.40^a^ ± 23.02	65.74^a^ ± 24.02	47.16^c^ ± 24.54				
Ash retention	Without skin	91.14^b^ ± 3.33	^x^95.89^a^ ± 2.51	66.62^c^ ± 2.14	84.53 ± 13.34	0.074	0.001	0.058
	With skin	88.90^a^ ± 3.27	^y^91.20^a^ ± 3.36	67.69^b^ ± 3.56	82.59 ± 11.28			
	Total	89.97^b^ ± 3.38	93.55^a^ ± 3.75	67.16^c^ ± 2.89				
Cholesterol retention	Without skin	^y^87.89^b^ ± 0.40	105.85^a^ ± 0.76	86.27^c^ ± 0.81	^y^93.34 ± 9.08	0.001	0.001	0.001
	With skin	^x^105.26^b^ ± 0.74	106.17^a^ ± 0.86	86.07^c^ ± 0.57	^x^99.17 ± 9.49			
	Total	96.58^b^ ± 8.99	106.01^a^ ± 0.80	86.17^c^ ± 0.68				

a–c: different letters in rows means statistically significant differences between group average, including thermal treatment (*p* ≤ 0.05); x–y: different letters in columns means statistically significant differences between group average, including kind of meat (*p* ≤ 0.05).

**Table 4 foods-12-00129-t004:** Mineral retention coefficients (%) in cooked goose meat (n = 6 breast muscles with skin and n = 6 without skin for each kind of heat treatment).

Retention (%)	Meat	Heat Treatment			Total	*p*-Value (*p* ≤ 0.05)		
		Microwave (M) Cooking	Sous Vide (SV) Cooking	Stewing (S)		Meat (Mt)	Heat Treatment (T)	Mt x T
Zinc (Zn)	Without skin	^x^152.15 ± 6.33	^x^148.36 ± 3.21	^x^151.98 ± 4.06	^x^150.83 ± 5.00	0.001	0.001	0.001
	With skin	^y^118.03^b^ ± 4.53	^y^131.70^a^ ± 7.81	^y^133.42^a^ ± 4.83	^y^127.72 ± 9.02			
	Total	135.09 ± 18.41	140.03 ± 10.36	142.70 ± 10.62				
Phosphorus (P)	Without skin	119.81^a^ ± 2.45	^x^119.70^a^ ± 2.10	^x^94.66^b^ ± 3.72	^x^111.39 ± 12.39	0.001	0.001	0.097
	With skin	118.46^a^ ± 4.84	^y^114.39^a^ ± 5.24	^y^91.06^b^ ± 2.64	^y^107.97 ± 13.03			
	Total	119.14^a^ ± 3.77	117.05^a^ ± 4.73	92.86^b^ ± 3.63				
Magnesium (Mg)	Without skin	^x^108.73^a^ ± 2.41	^x^109.39^a^ ± 3.23	^x^93.47^b^ ± 3.59	^x^103.86 ± 8.07	0.001	0.001	0.001
	With skin	^y^92.84^a^ ± 3.17	^y^86.47^b^ ± 3.54	^y^74.31^c^ ± 2.43	^y^84.54 ± 8.39			
	Total	100.79^a^ ± 8.64	93.43^a^ ± 12.28	83.89^c^ ± 10.33				
Sodium (Na)	Without skin	^y^34.04^a^ ± 0.54	^y^32.25^b^ ± 0.97	^y^25.20^c^ ± 1.22	^y^30.50 ± 4.00	0.001	0.001	0.001
	With skin	^x^44.57^a^ ± 2.03	^x^40.93^b^ ± 2.04	^x^32.21^c^ ± 0.86	^x^39.23 ± 5.55			
	Total	39.31^a^ ± 5.62	36.59^a^ ± 4.74	28.70^b^ ± 3.76				
Calcium (Ca)	Without skin	^x^88.83^b^ ± 2.82	^x^122.69^a^ ± 4.14	^x^80.21^c^ ± 1.95	^x^97.24 ± 18.96	0.001	0.001	0.001
	With skin	^y^81.16^b^ ± 3.02	^y^108.97^a^ ± 3.26	^y^65.81^c^ ± 2.69	^y^85.32 ± 18.47			
	Total	85.00^b^ ± 4.86	115.83^a^ ± 7.94	73.01^c^ ± 7.77				
Iron (Fe)	Without skin	^y^98.84^a^ ± 1.93	^y^81.03^c^ ± 2.09	^y^88.82^b^ ± 3.12	^y^89.56 ± 7.80	0.001	0.001	0.001
	With skin	^x^113.75^a^ ± 3.31	^x^97.22^b^ ± 3.74	^x^96.95^b^ ± 2.93	^x^102.6 ± 8.64			
	Total	106.30^a^ ± 8.13	89.13^b^ ± 8.86	92.89^b^ ± 5.12				
Potassium (K)	Without skin	^y^48.16^b^ ± 1.07	^y^58.38^a^ ± 1.49	^y^38.27^c^ ± 1.23	^y^48.27 ± 8.48	0.001	0.001	0.001
	With skin	^x^58.51^b^ ± 1.51	^x^62.16^a^ ± 2.55	^x^42.77^c^ ± 0.88	^x^54.48 ± 8.76			
	Total	53.34^b^ ± 5.49	60.27^a^ ± 2.81	40.52^c^ ± 2.55				

a–c: different letters in rows means statistically significant differences between group average, including thermal treatment (*p* ≤ 0.05); x–y: different letters in columns means statistically significant differences between group average, including kind of meat (*p* ≤ 0.05).

## Data Availability

The data can be made available at the request of the concerned persons by the contact person via e-mail.

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
