# Peer review of "The Effects of Sous Vide, Microwave Cooking, and Stewing on Some Quality Criteria of Goose Meat"

_foods, 2022, doi:10.3390/foods12010129_

Round 1
Reviewer 1 Report
foods-2063963The effect of sous-vide, microwave cooking and stewing on some quality criteria of goose meat
Dear Authors,
Your work is very interesting and it contributes significantly to the area of Meat Processing.
My only observations to your manuscript are:
Table 2 and 4. The column of “Raw Meat” could be deleted.
Line 467. Change “Contrary to our findings, in the Goluch et al. [7] studies,” by “Contrary to our findings, in the Goluch et al. [7] study,”
Line 547. Please delete space between digits and symbol in “23.8 %”
Line 561-590. This long discussion comparing your results with those of Goluch et al. [7) could be reduced. Please do it.
Line 604-616. Too long paragraph to describe a study of other authors. Please shorten it and discuss it straightaway with your findings.
Author Response
Thank You, dear Review, for time to read our article and prepare this very kind review.
Table 2 and 4. The column of “Raw Meat” could be deleted. - Done as requested
Line 467. Change “Contrary to our findings, in the Goluch et al. [7] studies,” by “Contrary to our findings, in the Goluch et al. [7] study,” - done as requested
Line 547. Please delete space between digits and symbol in “23.8 %” - Done as requested
Line 561-590. This long discussion comparing your results with those of Goluch et al. [7] could be reduced. Please do it. - Done as requested
Line 604-616. Too long paragraph to describe a study of other authors. Please shorten it and discuss it straightaway with your findings. - Done as requested
Reviewer 2 Report
Foods
foods-2063963
The effect of sous-vide, microwave cooking and stewing on some quality criteria of goose meat
Dear Editor,
The article deals with the evaluation of the influence of three cooking methods (sous-vide – S-V, microwave – M cooking, stewing – S) on basic chemical composition, cholesterol content, energy value, mineral concentration, and retention coefficients in goose meat. The topic is good. The manuscript has been well designed and written. My specific comments and questions are below;
- Lines 40 and 44: This sentence is the same in the abstract! Please rewrite it!
- Line 88: Why were female geese used in this research?
- What was the internal temperature for stewed samples?
- Lines 144 and 145: 12 h or to constant weight?
- “Moisture content” should be “water content”. Because the water contents of the samples were higher than 50%.
Author Response
Thank You, dear Review, for time to read our article and prepare this very kind review. I hope that all comments were fully taken into account and explained.
Lines 40 and 44: This sentence is the same in the abstract! Please rewrite it! - done as requested
Line 88: Why were female geese used in this research? Goose (female) muscles were used for the study as only female geese are available for sale.
What was the internal temperature for stewed samples? The final cooking temperature (75℃) in all of thermal treatment was monitored in the geometric centre of each muscle.
Lines 144 and 145: 12 h or to constant weight? The weight of the muscles used for sous-vide processing was similar, with a deviation of 35 grams for samples without skin and 48 grams for samples with skin. The values ​​of deviations did not exceed 10% of the weight of the analyzed muscles.
“Moisture content” should be “water content”. Because the water contents of the samples were higher than 50%. done as requested
Reviewer 3 Report
1. The title may be a little modified (Quality criteria word may be changed with any suitable alternative)
2. Introduction (Typing error)
3. The reason behind increased protein values in different cooking methods needs a better explanation.
4. Please explain the reason behind increase in energy value of cooked goose breast meat with skin in microwave cooking as compared to S-V and S methods.
5. When the M & S cooked meat were analyzed as such, why the S-V cooked meat was refrigerated for 24 hours? Similar storage conditions would have given better results.
6. Why oven-dried samples were used for proximate composition assay, but freeze-dried samples were used for mineral analysis? The drying methods too affect the meat composition.
7. The raw samples with skin had higher (1.01 MJ/100g) energy value compared to cooked meat (0.90-0.95 MJ/100g). The heat processing increased skin-less breast muscles' energy value, but why not in the case of cooked goose meat with skin? (As we know, the energy levels of samples correspond to their fat contents, and thus meat containing skin/fat must have higher energy levels than those without.)
8. The cholesterol content in meat mostly decreases due to heat treatment as cholesterol is heat-labile. A suitable explanation may be given for the increased cholesterol content in goose meat after cooking.
9. The reasons behind the increase and decrease of individual minerals in cooked meat (skinless & with skin) must be explained.
10. Sensory evaluation of cooked goose meat (skinless & with skin) by specialized panel members after different cooking methods will give better clarity on consumer acceptability. The incorporation of sensory evaluation data will practically serve the purpose of this study.
This study has mostly discussed the similarity of findings with other studies. The possible scientific reasons behind the observed results should be explained properly.
The observed variations may not be only due to cooking methods. The individual variations should have been ruled out. Furthermore, the work-study is very simple and there is not much novelty in this work. However, the incorporation of organoleptic evaluation data will practically be beneficial from the consumer’s perspective.
Author Response
Thank You, dear Review, for time to read our article and prepare this very insightful review. I hope that all comments were fully taken into account and explained.
- The title may be a little modified (Quality criteria word may be changed with any suitable alternative). Thank You for suggestion, but the authors of manusript consider that the more autoritative word of the phrase is quality criteria. In addition, the publication is financed from Interekon program and the title of this publication was submitted when I submitting the application for funding.
- Introduction (Typing error) - thank You, it has been corrected.
- The reason behind increased protein values in different cooking methods needs a better explanation - done as requested, the information has been added in text.
- Please explain the reason behind increase in energy value of cooked goose breast meat with skin in microwave cooking as compared to S-V and S methods. - These differences may resulted from different fat content. These differences, confirmed statistically, from the nutritional point of view, were not significant (only 0.04 and 0.05 MJ/100g sample).
- When the M & S cooked meat were analyzed as such, why the S-V cooked meat was refrigerated for 24 hours? Similar storage conditions would have given better results. - All heat-treated samples were refrigerated for 24 hours before the analysis. In line 105-106 we wroten: “After heat processing, muscles were allowed to cool to room temperature. Then, the muscles were stored at 4℃ for 24 h in the refrigerator”.
- Why oven-dried samples were used for proximate composition assay, but freeze-dried samples were used for mineral analysis? The drying methods too affect the meat composition. -
This is the standard methodology applied to raw meat provided by AOAC. To determine ash, temperatures above 500°C must be used, and the only method used to determine ash, whether in meat or vegetable raw material, is combustion in a muffle furnace. On the other hand, lyophilization is a method that has the least impact on the raw material and can be used as an initial method of preparing samples for analysis.
- The raw samples with skin had higher (1.01 MJ/100g) energy value compared to cooked meat (0.90-0.95 MJ/100g). The heat processing increased skin-less breast muscles' energy value, but why not in the case of cooked goose meat with skin? (As we know, the energy levels of samples correspond to their fat contents, and thus meat containing skin/fat must have higher energy levels than those without.) - In the scientific literature, there is no information on the effect of thermal treatment on changes in the basic chemical composition, taking into account muscles with and without skin. Our work is, to my knowledge, one of the first on this subject. Burrowing poultry does not have significant amounts of subcutaneous fat. Water poultry, due to its environmental conditions, contains significant amounts of subcutaneous fat. It can be assumed that the increase in the fat content in the skinless muscles, and thus the energy value (calculated on the basis of the fat and protein content in the analyzed goose muscles) may be caused by water leakage. In the case of muscles analyzed without skin, we are dealing with the analysis of fat located in the muscle, i.e. intramuscular fat. In the case of muscles analyzed with the skin, we analyze the fat located in the muscles and subcutaneous fat, which is more easily rendered. In muscles without skin, water loss results in an increase in fat content, and in muscles with skin, there are more variables affecting the results of leakage and fat content.
- The cholesterol content in meat mostly decreases due to heat treatment as cholesterol is heat-labile. A suitable explanation may be given for the increased cholesterol content in goose meat after cooking. - Thank You for suggestion, the explanation was added to text.
- The reasons behind the increase and decrease of individual minerals in cooked meat (skinless & with skin) must be explained. -
Thank You for suggestion, this informations were added in text.
- Sensory evaluation of cooked goose meat (skinless & with skin) by specialized panel members after different cooking methods will give better clarity on consumer acceptability. The incorporation of sensory evaluation data will practically serve the purpose of this study. -
Thank You for suggestion. The sensory evaluation of cooked goose meat (with skin and without skin; cooking metohods: sous-vide, microwave cooking, and stewing) carried out by specialized members panel were done. This article about selected physical properties of goose meat such as sensory evaluation, cooking loss, texture, color parameters in now in review.
Round 2
Reviewer 1 Report
Dear Authors,
Thank you for the corrected version of your manuscript.
No further changes are needed.
Author Response
Thank You,
Dear Rewiever,
Authors
Reviewer 3 Report
The author has incorporated all the corrections
Author Response
Thank You, Dear Rewiecer,
Authors